# "You Are An Expert Linguistic Annotator": Limits of LLMs as Analyzers of Abstract Meaning Representation

**Allyson Ettinger\*[1], Jena D. Hwang\*[1],**
**Valentina Pyatkin[1], Chandra Bhagavatula[1], Yejin Choi[1,2]**
[1]Allen Institute for AI  [2]University of Washington
{allysone,jenah,valentinap,chandrab,yejinc}@allenai.org

## Abstract

Large language models (LLMs) show amazing proficiency and fluency in the use of language. Does this mean that they have also acquired insightful linguistic knowledge *about* the language, to an extent that they can serve as an "expert linguistic annotator"? In this paper, we examine the successes and limitations of the GPT-3, ChatGPT, and GPT-4 models in analysis of sentence meaning structure, focusing on the Abstract Meaning Representation (AMR; Banarescu et al. 2013) parsing formalism, which provides rich graphical representations of sentence meaning structure while abstracting away from surface forms. We compare models' analysis of this semantic structure across two settings: 1) direct production of AMR parses based on zero- and few-shot prompts, and 2) indirect partial reconstruction of AMR via metalinguistic natural language queries (e.g., *"Identify the primary event of this sentence, and the predicate corresponding to that event."*). Across these settings, we find that models can reliably reproduce the basic format of AMR, and can often capture core event, argument, and modifier structure—however, model outputs are prone to frequent and major errors, and holistic analysis of parse acceptability shows that even with few-shot demonstrations, models have virtually 0% success in producing fully accurate parses. Eliciting natural language responses produces similar patterns of errors. Overall, our findings indicate that these models out-of-the-box can capture aspects of semantic structure, but there remain key limitations in their ability to support fully accurate semantic analyses or parses.

## 1 Introduction

LLMs in recent years have revolutionized artificial intelligence, showing advanced proficiency and fluency in the use of language, and appearing to possess high levels of expertise and analytical capability across a wide variety of specialized domains.

---
*Equal contribution

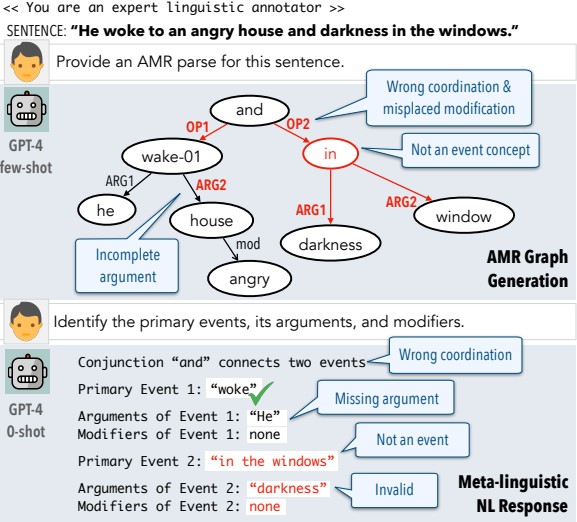

Figure 1: We compare LLMs' ability to generate structured semantic information across two settings: zero- and few-shot generation of AMR parses, and metalinguistic natural language queries.

Observation of these capabilities has raised important questions about the extent, robustness, and limitations of the knowledge and analysis abilities of these models in specialized domains.

In this paper we zero in on the domain of linguistic analysis: these models have shown great proficiency with language, but here we ask **not just how well the models *use* language, but how much they know *about* language.** Specifically, we explore to what extent models are able to analyze the meaning of a sentence and reproduce the structure of that meaning. Most directly this allows us to conduct a status check on the level of expertise that LLMs have acquired in linguistic analysis, and to assess to what extent linguistic structural annotation can be done reliably by LLMs out of the box. At a higher level, this investigation has potential implications for the robustness of models' abstract representation of meaning in language inputs. We intend for this to serve as a brief status report with respect to model capabilities in this domain.

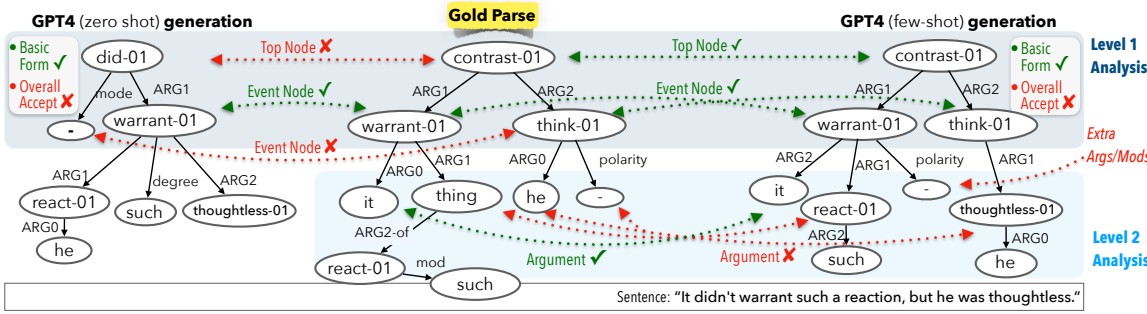

Figure 2: We use a two-tiered semantic evaluation framework to gain fine-grained insight into the strengths and limitations of generated parses. In the example, GPT-4 parse generations are compared against the gold parse.

For examining models' capability in analysing linguistic meaning structure, we focus on a case study of the Abstract Meaning Representation formalism (AMR) (Banarescu et al., 2013). AMR is designed to capture the abstract structure of sentence meaning, disentangling this structure from surface forms of language. It formalizes semantic structure of a sentence into directed graphs that capture "who did what to whom" as well as detailed abstract information on how aspects of the sentence meaning modify and relate to one another.

In our explorations, we examine models' ability to produce the structural meaning information contained in AMR parses across three settings: zero-shot generation of AMR parses, few-shot generation of AMR parses, and zero-shot generation of natural language descriptions. We test GPT-3, Chat-GPT, and GPT-4. Our results show that all models are able to reproduce the basic AMR format and structure, and they can in principle produce correct outputs at any level of AMR—with greatest reliability on core event-argument triplets corresponding to subject-verb-object structures. However, models are prone to frequent and major errors in capturing the semantic structure (see Fig. 1, 2), and when we assess the parses for overall acceptability, we see virtually 0% success rate across models. Comparisons between patterns in parse and natural language output settings suggest that these limitations are not simply artifacts of the output type, and may reflect more fundamental limitations in models' capacity for semantic analysis. Overall, our findings indicate that although models can execute impressively formatted and partially correct semantic parse outputs, the prevalence of errors outside of basic components is such that these models cannot be used reliably out-of-the-box for generating this type of structured abstract meaning information, and more involved techniques are needed to adapt these models effectively for such purposes.

## 2 Related Work

A large body of work has examined various aspects of syntactic and semantic capabilities in language models (c.f. Mahowald et al., 2023), indicating that LLMs show strong knowledge of syntactic structure, while semantic capabilities are more mixed. Nonetheless, LLMs have also been used for few-shot semantic parsing with some success. In particular, Shin et al. (2021) and Shin and Van Durme (2022) find that few-shot learning in GPT-3 and Codex produces semantic parses that outperform baselines with comparable training sizes. These semantic parsing datasets, which focus on producing database queries in particular domains, are less complex and domain-general than AMR, but the results suggest that LLMs should contain aspects of the knowledge needed to analyze semantic structure. As for AMR, pre-trained transformer models have helped to advance the state of the art in AMR parsing, with recent AMR parsers building on the foundation of models like BART (Bevilacqua et al., 2021; Bai et al., 2022; Lee et al., 2022; Zhou et al., 2021). This indicates that pre-trained models may also pick up on representational capabilities relevant for supporting AMR.

Though these prior works are suggestive that LLMs and pre-trained transformers capture certain aspects of linguistic structure, it is not clear from existing results how detailed or reliable LLMs' ability to analyze meaning structure may be—formalisms used for prior few-shot semantic parsing are simpler and more domain-specific than AMR, and the supervised fine-tuning of BART for AMR parsing obscures the details of what original knowledge may have been contained in the pre-trained model. To achieve a clearer picture of LLMs' ability to analyze rich semantic structure,

| **Level 1 criteria** | |
|---|---|
| *Proportion of parse outputs that...* | |
| Basic Form | meet basic AMR graph format (concept nodes, edge relationships, hierarchical bracketing) |
| Top Node | correctly identify the top parse node |
| Main Rel | select the correct main *event* as the highest event relation, disregarding non-eventive relations like *and* (Fig. 1) or *contrast-01* (Fig. 2) |
| Overall Accept | constitute a valid AMR representation for the sentence, regardless of match to the gold annotation (allows us to credit models if they parse the sentence reasonably but differ from gold) |
| **Level 2 criteria** | |
| *Proportion of parse outputs in which, for each correctly identified top-level event, ...* | |
| Event Args | all arguments are present and identified as arguments of that event |
| Event Mods | all event modifiers are present and identified as modifiers of that event |
| Arg Mods | all modifiers for arguments of that event are present and identified as modifiers of the args |
| Extra Mods | there are any modifiers added erroneously to events or arguments anywhere in the parse |

Table 1: Semantic criteria used for analysis

we directly examine pre-trained models' ability to produce AMR information, and we do so across a number of potentially productive zero- and few-shot settings for maximum insight about model capabilities. We also prioritize fine-grained, manual analysis of models' accuracies at multiple levels of AMR information, in order to provide more detailed insights into model capabilities.

## 3 Evaluation

### 3.1 Evaluation Framework

Standard metrics for evaluating AMR include Smatch and SemBLEU, which provide holistic analysis of node matches between generated and gold AMR parses. While these metrics are well-suited for large-scale quantitative evaluation, they are not adequate for detailed understanding of models' strengths and limitations in capturing AMR information. For more detailed insight, we lay out a novel fine-grained evaluation framework. We define two levels: **Level 1** criteria to capture basic format, highest-level nodes, and overall semantic accuracy; and **Level 2** criteria for assessing accuracy with arguments and modifiers. Table 1 outlines the analysis criteria (further details in §C).

### 3.2 Data

To ensure maximum flexibility and expert-level accuracy in assessment of the above criteria, we carry out our evaluation manually. In choosing to use fine-grained manual evaluation, we necessarily accept a tradeoff with respect to scale and generalization guarantees, as expert manual evaluation is time-consuming. We are not the first to accept this tradeoff: due to cost and increasing complexity of LLM outputs, there is increasing precedent for analyzing model capabilities even on samples of single outputs (e.g., Bubeck et al., 2023). Here we seek a balance between this kind of single-instance analysis and larger-scale coarse-grained evaluation, via fine-grained manual analysis on a small exploratory test set sampled across several domains.

To this end, we compile a sample of 30 AMR gold-parsed sentences, randomly selecting 10 sentences of varying character lengths from the gold AMR annotated AMR 3.0 (**AMR3**; Knight et al. 2021) and Little Prince[1] (**LPP**) datasets, and also sampling and annotating 10 sentences from websites published in 2023 (**2023**), to test the possibility of memorization of public AMR annotations available in pre-training. This is a large enough sample to gain some insight into trends at our different levels of analysis, and future studies at larger scale can provide further insight into patterns that emerge in larger samples. See §A for more details.

## 4 Zero-shot AMR parsing

Given findings of superior zero-shot performance in a wide variety of domains (Bubeck et al., 2023), we begin by testing models' zero-shot capability for generating AMR graphs directly. Instructions and examples for AMR annotation are widely available online, so it is reasonable to imagine that models may learn to do this task zero-shot as well.

We input to the model the target sentence and the simple instruction "Provide an AMR (Abstract Meaning Representation) parse for this sentence." For ChatGPT and GPT-4, we include the system message "You are an expert linguistic annotator." Our goal here is to use clear and fair prompts that allow us to assess model capabilities and limitations. We do not do elaborate prompt engineering, but take the stance that if a prompt is sufficiently clear, then a failure to perform the task is simply a failure—reliance on particular prompt phrasing or structure is an indication of model brittleness.

---

[1] https://amr.isi.edu/download.html

| | Model | Basic Form | Main Rel | Top Node | Accept |
|---|---|---|---|---|---|
| 0-shot | GPT-3 | 0.7 | 0.4 | 0.3 | 0.0 |
| | ChatGPT | 0.7 | 0.4 | 0.2 | 0.0 |
| | GPT-4 | 1.0 | 0.4 | 0.4 | 0.0 |
| 5-shot | GPT-3 | 1.0 | 0.7 | 0.5 | 0.0 |
| | ChatGPT | 1.0 | 0.8 | 0.5 | 0.0 |
| | GPT-4 | 1.0 | 0.7 | 0.6 | 0.1 |

Table 2: Results of Level 1 analysis of zero- and few-shot parses on the 30 selected sentences

| | Model | Event Args | Event Mods | Arg Mods | Extra Mods (↓) |
|---|---|---|---|---|---|
| AMR (5-shot) | GPT-3 | 0.4 | 0.4 | 0.1 | 0.4 |
| | ChatGPT | 0.4 | 0.5 | 0.2 | 0.5 |
| | GPT-4 | 0.5 | 0.5 | 0.4 | 0.4 |
| Meta-linguistic | GPT-3 | 0.5 | 0.3 | 0.2 | 0.4 |
| | ChatGPT | 0.6 | 0.4 | 0.2 | 0.3 |
| | GPT-4 | 0.6 | 0.4 | 0.4 | 0.3 |

Table 3: Level 2 analysis of few-shot and NL outputs

Since the zero-shot parses fail often even at the basic levels, we limit to Level 1 analysis for this setting. Results are in the top segment of Table 2. The outputs indicate clearly that these LLMs have been exposed to AMR parse annotations in their pre-training, and have managed to learn surface characteristics of AMR structure: we see that for all models a majority of outputs (>70%) show basic AMR format despite the absence of any demonstration in the prompt. Comparison between publicly annotated sentences and newly annotated sentences from 2023 (§B) also shows no noteworthy difference, indicating that output quality is not reliant on presence of test AMR annotations in pre-training.

However, beyond the basic form, all models show frequent and substantial errors in the parsing. The parses identify the correct top node only 20-40% of the time, reflecting routine failures to incorporate clausal and discourse relations that AMR often captures in the top node—and even with the more relaxed criterion of identifying the main event relation, LLMs succeed in only about half of the parses (40%). When we consider the viability of the full structure as an appropriate meaning representation of the sentence, none of the models produce any fully acceptable AMR parse. These results suggest that out-of-the-box, zero-shot LLM capabilities are limited primarily to mimicking surface format of AMR representations, with understanding of the linguistic functions and phenomena being beyond their zero-shot capabilities.

## 5 Parsing with few-shot demonstrations

Given that zero-shot parsing shows non-trivial limitations across all models, we next test how parses improve with few-shot demonstrations of AMR parsing. We use the same instructions (and, in ChatGPT and GPT-4, system message), but we now include the specification "I will first show some examples." followed by five example sentences with corresponding AMR parses, selected based on sim-

ilarity to the test sentence.[2]

For few-shot parses, we apply both Level 1 (Table 2) and Level 2 (Table 3) evaluations. We see that all parses now conform to AMR format, and the main event is now correct a majority of the time. Identification of the top node has also improved, with correct outputs in approximately half of cases. However, the percentage of overall parse acceptability has made virtually no improvement, despite the explicit few-shot demonstrations.

For Level 2 analysis, we see that models have limited reliability in identifying a given event's arguments and modifiers (40-50%) or argument modifiers (10-40%). Additionally, just under half of parses include at least one spuriously-identified argument or modifier ("Extra Mods"). Qualitative analysis indicates that models make diverse errors that can occur at any level of AMR structure, though they show the most reliable accuracy in representing event-argument triplets corresponding to subject-verb-object structures. See §E for additional examples and discussion on these points.

## 6 Metalinguistic NL responses

By far the most thoroughly trained format for LLMs is that of natural language, so we next explore models' ability to use natural language to identify and describe the abstract meaning structure relevant for AMR, via prompting for metalinguistic information about the target sentence. To do this, we formulate a natural language prompt instructing the model to identify and break down aspects of the sentence meaning structure corresponding to components of AMR, similar to the process that an AMR annotator would use. Our prompt for this setting is shown in §D.

In this setting, the prompt asks for a breakdown of events, arguments, and event/argument modifiers, but does not elicit enough information to enable complete parses. For this reason, we focus

---

[2]Sentence similarity is computed via Universal Sentence Encoder embeddings (Cer et al., 2018).

on our Level 2 criteria for analyzing these outputs. Results are shown in the bottom of Table 3. We see that the overall patterns of accuracy are strikingly similar to those in the few-shot case (with marginal differences that are too small to read into with these small samples).[3] This suggests that limitations in the zero- and few-shot parses are not due simply to difficulty in generating parse format, but may reflect more fundamental limitations in models' current capacity to analyze semantic structure in language. This conclusion is further supported by the observation that the instructions contained in the metalinguistic prompt do not appear at any level to be fundamentally too difficult for models to interpret: though models make many errors, for every component of the prompt they show in at least some cases the ability to produce correct outputs for that component. See §E for example outputs in the NL setting and discussion of instruction-following successes and error patterns.

**Comparing parse vs NL output** Side-by-side inspection of parse and NL outputs supports the conclusion that, although errors are not identical, the two output formats may reflect real patterns in models' analytical capabilities. An illustration can be seen in Figure 1, which shows GPT-4 few-shot parse and zero-shot NL outputs for the sentence *"He woke to an angry house and darkness in the windows"*. This is a simple sentence, but the argument structure of the verb *woke* lacks the simple subject/object structure that models succeed at most often—and perhaps for this reason in both output formats the model misinterprets the sentence to include two separate events (missing the fact that the *angry house and darkness in the windows* is a single argument of the *waking* event) and creating nonsensical argument and modifier structures as a result of the mistaken analysis. Additional side-by-side examples are included in Figures 5, 6, and 7 in the Appendix, further illustrating similarities between parse and NL outputs, and supporting the possibility that the observed errors in these outputs reflect real underlying analytical limitations rather than artifacts of instructions or output format.

## 7 Smatch comparison

To anchor our results relative to an existing metric, we obtain Smatch scores for our five-shot GPT-4

|  | AMR3 | 2023 | LPP |
|---|---|---|---|
| AMRBart | 0.78 | 0.79 | 0.75 |
| GPT-4 (No fixes) | 0.42 | 0.26 | 0.42 |
| + Auto fixes | 0.48 | 0.40 | 0.47 |
| + Manual fixes | 0.51 | 0.47 | 0.51 |

Table 4: AMRBart and GPT-4 few-shot smatch-score

parses and compare against those for the supervised AMRBART parser (Bai et al., 2022) on the same test sentences. Since GPT-4's generated parses are often flawed to the point of Smatch not being able to run, we report three methods for obtaining these scores: **no fixes**, in which all failing parses are simply replaced with single-node placeholder parses; **auto fixes**, in which some automated format fixes are applied (see §F for details) and remaining errors are replaced with placeholders; and **manual fixes**, in which we supplement automatic fixes with manual fixes to correct remaining format errors. The Smatch results (Table 4) clearly show that GPT-4 output quality is far below that of the supervised AMR parser, supporting our general observation that the quality of these LLM parses is limited.

## 8 Conclusion

Our analyses show that LLMs have acquired sufficient knowledge of AMR parsing and semantic structure for reliable generation of basic AMR format and partially correct representations of sentence meaning. However, we see abundant, diverse errors in model outputs, virtually no fully accurate parses, and error patterns suggesting real underlying limitations in models' capacity to analyze language meaning. Our findings indicate that models are not currently sufficient out-of-the-box to yield reliable and accurate analyses of abstract meaning structure, and overall that this is a domain in which models show only mixed levels of expertise.

We are confident that additional fiddling and clever manipulations can further improve the outputs of these models, at least on certain dimensions. However, we present these results as a current status report and reality check to counterbalance frequent claims focused on widespread success and intelligence of these models out-of-the-box. We look forward to continuing work to better understand the fundamental strengths and limitations of these models in this domain, and to improve the reliability of semantic analysis capabilities achievable through collaboration with these models.

---

[3]Though not included in Table 3, we note that the NL outputs also show comparable accuracy to the few-shot parses in the Main Rel category for all models.

## Limitations

In this paper we intend to provide an overview and status check on the out-of-the-box capabilities of current LLMs for the rich semantic analysis captured by AMR parses. To enable fine-grained manual evaluation not possible through standard metrics, we have used a small exploratory test set, and consequently our results do not enable statistical comparisons or claims about how patterns may play out at larger scale. We look forward to future work applying comparable fine-grained analysis on larger samples, to verify what additional patterns of success and failure may emerge, and what broader generalizations can be made about model capabilities in this domain.

A potentially valuable extension that we do not include here would be a detailed comparison with models' success in other (likely simpler) semantic or syntactic parsing formalisms. Given the richness of AMR and our focus on abstract semantic structure per se, we do not include such an analysis in the current work.

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

## A Dataset Details

### A.1 More on Abstract Meaning Representation

AMR formalizes semantic structure of a sentence into directed graphs that capture the "who did what to whom" of the sentence (Banarescu et al., 2013). In AMR, events and entities are represented as **concept** nodes, and semantic relationships or **semantic roles** as edges. AMR abstracts away from syntactic and morphological surface variations in favor of conceptual representation of predicate-argument structure of a sentence in part by adopting English PropBank (Kingsbury and Palmer, 2003) for event representation. In this way, AMR allows meaning generalization across various surface expressions (e.g., "The girl adjusted the machine" and "The girl made adjustments to the machine" would have the same AMR graph). AMR supports other linguistic analysis such as coreference, named entities, and modality (among others).

### A.2 Data Sources

**AMR 3.0.** We use the AMR 3.0 dataset (**AMR3**; LDC2020T02), which includes 59K AMR graphs. The graphs are manually annotated from English natural language sentences from various genres of text including newswire, discussion forums, fiction and web text. We particularly focus on two English subsets: BOLT discussion forum data and LORELEI data. Our few-shot in-context examples and test instances are pulled from the AMR 3.0 training and dev sets, respectively.

**The Little Prince.** We use publicly available AMR annotations of the novel *The Little Prince* by Antoine de Saint-Exupéry (translation of original French, *Le Petit Prince*; **LPP**). The corpus contains 1.5K sentences with their corresponding, manually-created AMR graphs. Our few-shot in-context examples and test instances are non-overlapping samples drawn from this dataset.

**2023 Sentences.** To experiment with sentences verifiably not present in pre-training, we randomly sample sentences from websites published in 2023. To obtain the AMR gold parses (2023), we run the Spring Parser (Bevilacqua et al., 2021) on the sentences, and then the output is manually corrected by one of the authors with expertise in AMR annotation. These 2023 sentences are used in the test set, and few-shot examples for the 2023 sentences are drawn from the AMR 3.0 training set.

### A.3 Test Data Selection

AMR3 data was sourced from the AMR 3.0 BOLT and LORELEI instances with publicly available unified annotation from PropBank (Bonial et al., 2014).[4] The Little Prince has also been partially annotated with Universal Dependencies (Nivre et al., 2020) parses (Schneider et al., 2018), and for this work we sourced from those The Little Prince instances that had both AMR and Universal Dependencies annotation. In both cases, we selected sentences of 40-300 character length to eliminate incomplete phrases as well as overly long sentences, producing 413 AMR 3.0 and 67 The Little Prince instances. We then narrowed the sets to 10 random instances from each of the two data subsets.[5] For 2023 Sentences, we manually selected sentences from online news sources and blogs with article date stamps of January 2023 or later. We selected 30 sentences, then narrowed the set to 10 instances of varying character lengths.

## B Concerns of Memorization

We took into consideration the possibility that the parses of AMR3 and LLP could be present in the pre-training data of the tested LLMs. This served as the motivation for including the 2023 sentences.

Table 5 shows Level 1 zero-shot parse results broken down by dataset. These results suggest that the quality of the AMR generations is not reliant on direct memorization of the annotated parses from pre-training—in fact, we find the results for 2023 parses to be nearly identical to the LPP results. A closer qualitative look at the parses did not surface any noteworthy differences in parses.

---

[4] https://github.com/propbank/propbank-release
[5] We ensured inclusion of diverse lengths via manual verification.

| | Model | Basic Form | Main Rel | Top Node | Accept |
|---|---|---|---|---|---|
| AMR3 | GPT-3 | 0.8 | 0.5 | 0.3 | 0 |
| | ChatGPT | 0.7 | 0.4 | 0.3 | 0 |
| | GPT-4 | 1 | 0.6 | 0.5 | 0 |
| LPP | GPT-3 | 0.6 | 0.3 | 0.2 | 0 |
| | ChatGPT | 0.8 | 0.2 | 0.1 | 0 |
| | GPT-4 | 1 | 0.3 | 0.4 | 0 |
| 2023 | GPT-3 | 0.6 | 0.3 | 0.3 | 0 |
| | ChatGPT | 0.9 | 0.2 | 0.3 | 0 |
| | GPT-4 | 1 | 0.3 | 0.4 | 0 |

Table 5: Comparison of the Level 1 analyisis on zero-shot data across the three datasets

## C Analysis Criteria Details

Here we provide further elaboration and illustration with respect to the analysis criteria outlined in Table 1.

**Assessment of Basic Form** For the Basic Form criterion we simply ask whether or not the produced output looks (generally) consistent with AMR's standard structure. More specifically, a parse should critically retain three basic AMR format components: concept nodes, edge relationships (whether *ARG#* or modifier), and hierarchical bracketing notation. We do not require the parses to contain variables (e.g. *b* in *(b / boy)*) or to use rounded parentheses (e.g. *( )*), but we find that every generation with the three critical components also includes variables and parentheses. An example of a parse that receives a zero for the Basic Form is given in Figure 3. Figure 4 shows a parse that is not fully up to AMR standards (e.g. *default-01~e.0* is not a standard format for concept representation) but that we credit for retaining the Basic Form.

Figure 3: ChatGPT output receiving Basic Form score of 0

Figure 4: ChatGPT output that is not fully up to AMR standards but passes our Basic Form criteria

**Overall acceptability** For our overall acceptability measure, an AMR expert among the authors assessed whether each parse could be a valid representation of the sentence meaning, based on the AMR annotation guidelines, regardless of match to the gold annotation. This was intended to give fairer credit to model outputs—none of the models' generated parses managed to perfectly match the gold annotation, but it was possible that some parses may still accurately represent the meaning of the sentence, with some annotation differences from the gold parse. So this measure used expert assessment to judge parse validity in this broader sense. These assessments also forgave minor structural/formatting errors, as long as a correct semantics could be interpreted.

**Main Rel vs Top Node** At times the main event relation in a parse will also be the top node, but in AMR non-eventive relations (e.g., discourse markers, conjunctions, modality) can instead take the top node position (e.g. *warrant* in Figure 2 is below the top node *contrast-01* representing *and*). Main Rel disregards whether models can recognize these non-eventive relations, and focuses instead on models' ability simply to recognize the main event of the sentence.

**Relaxation on exact match** For all of our match-based criteria, we evaluate based on *relaxed* matches rather than exact match. For concept nodes, we ignore PropBank sense labels and inflectional variations, and allow matching based on synonyms or otherwise differently-realized versions of the target concept. For example, *serve-01* is considered a match to *serve*, *served*, or *serve-02*—and although the AMR gold parse for "She served as a president for ...") uses *have-organization-role-91*

for "serve" (the standard AMR method for annotating organizational role, occupation, or profession) we also give credit to generated nodes labeled as any variant of *serve*.

For edge match, the only critical distinction is that between arguments (e.g., *ARG0*, *ARG1*)[6] and modifiers (e.g., *:time*, *:purpose*). Models receive credit for identifying an argument as *ARG* even if the number is mismatched, and similarly receive credit for identifying a modifier as a modifier without regard to the semantic specificity.

We also relax exact match on AMR's named entity types (e.g., node concept *organization* in (o / organization :name (n / name :op1 "Morgan" :op2 "Stanley"))). So long as the node concept is a reasonable match (e.g., *company* vs. *organization*), the models receive credit.

This use of relaxed match increases the need for expert manual annotation, but allows us to credit general semantic competence beyond match to specific AMR conventions.

## D  Metalinguistic prompt

We use a single instruction prompt for the metalinguistic natural language output setting reported in §6. The prompt used is shown below:

> (*System*: You are an expert linguistic annotator.)
> Sentence: *<replaced with input sentence>*
>
> Identify the primary event of this sentence, and the predicate corresponding to that event. If there are multiple equally primary events connected by a conjunction like "and", identify the conjunction, and then identify each of the primary events and their corresponding predicates.
>
> For each primary event, identify the arguments of the event predicate, and identify the modifiers of those arguments.
>
> Then for each primary event, identify any additional modifiers of that event.

## E  Example outputs and discussion

In this section we illustrate with additional examples some of the successes, failures, and overall patterns in the LLM outputs. Figures 5–7 show representative example outputs from GPT-4 in both parse generation (few-shot) and NL response settings (and an additional NL output example is in Figure 8). We highlight a number of points.

**Instruction-following success**  For the NL output setting in particular, the prompt instructions are somewhat complex, so it is worth considering

whether the instructions are too difficult for models to map to correct outputs. However, examination of successes across model generations indicates that no part of the NL setting instructions is fundamentally too difficult for the models to interpret and respond to. In the NL response setting shown in Figure 5b, the model is able to identify the primary event and arguments, and sort through and label modifiers for both the arguments and the event. Similar competence can be seen in the parse generation setting (Figure 5a): GPT-4 correctly identifies the main event selection and major arguments and modifiers. For conjunctions between events, we see in Figure 6 that in both NL and parse settings the model is able to handle the central conjunction "and", and break down the two coordinated components accordingly—even breaking down the second coordinate into its own two component sub-events. On this basis we can have reasonable confidence that the prompts are sufficiently clear and interpretable for the models.

**Errors are abundant and diverse**  Though models show the capacity in principle to handle any component of AMR information, examination of generated outputs shows that errors are abundant, diverse, and observable at every level of AMR structure. In addition to the illustration of output errors in Figure 1 and Figure 2, we see that even in the largely successful example in Figure 6, the model has misidentified the primary event in the first coordinate—the main point should be that *there was no issue*, not that the speaker *boarded the train*. Further, in the NL response, it has mistakenly identified "next to us" as a modifier of the *luggage*, rather than an argument of the *put* event.

In Figure 7 we see that the model is unable to identify the main event in either the parse generation or the NL setting. Though the main event relation is most appropriately identified as "imagine", in the generated parse the event "amaze" rises as the top event, and in the NL response output, "awaken" is identified as the main event. Even if we ignore the non-eventive information captured by *cause* (arising from the discourse marker "thus") and *possible-01* (signalled by the modal "can") concepts, the model fails to show sensitivity to the fact that "imagine my amazement" conveys the central information through which the content in the rest of the sentence is introduced.

Finally, Figure 8 shows another more extreme failure in the NL response setting. Here the model

---

[6]We simply treat *ARG#-of*s as *ARG*s.

has mistakenly zeroed in on "will look wonderful" as the primary event, rather than "it is a standout piece", and as a result it has defined arguments and modifiers in a variety of nonsensical ways.

**Most reliable with core event triplets**    Though errors are diverse and fairly idiosyncratic, one trend that emerges is that models show the most reliable performance with core event-argument triplets for individual verbs, most often corresponding to subject-verb-object triplets. For example, in Figure 5, in both formats the model clearly identifies the event "churn" and its arguments "the K-pop music sphere" and "newest catchy songs". In Figure 6, in both formats the model correctly structures the event "board" with its arguments "we" and "train", and the event "put" with its arguments "we" and "the luggage". In Figure 7, in both formats the model correctly captures the event "awaken" with its arguments "I" and "an odd little voice". This suggests that the model has a solid grasp on core verb argument structure—or at least that corresponding to subject-verb-object triplets—and can reliably map this to AMR form. However, beyond this core event structure, model performance becomes substantially less reliable.

**Parallel patterns between parse and NL**    We observe in Section 6 that there are similarities in the basic patterns of success and failure across LLM parse and NL outputs, and we highlight Figure 1 as an example. We see these parallels in Figures 5–7 as well: for instance, as we have just discussed, the consistent success on verb-argument triplets described above is seen in both parse and NL outputs for each example. More broadly, in Figure 5 we see that in both formats the model is successful on nearly the full AMR structure: it identifies the main verb ("churn") and its arguments ("sphere" and "song") and modifiers ("vie for"), and it captures semantic modifiers for the arguments in a coherent manner (e.g., "K-pop" is recognized as modifier of "music sphere"). In Figure 6, in both settings the model is successful in identifying "and" as a top-level conjunction joining two events, but makes the error of choosing "boarding a train" as the main event in the first coordinate of the structure. Similarly, in Figure 7 both output settings capture core event structure of "awaken", but miss out on the central event "imagine".

There are certainly divergences in output errors between different output formats for a given sentence. However, these divergences often stem from the fact that the tasks in these two settings do differ. For example, errors like the missing *have-degree-91* in Figure 5a—an AMR device meant to structure information extent—is not possible in the NL setting, as this level of structured detail is not requested in the prompt. Similarly, "sit down" in the parse in Figure 6a is split into two concept nodes, but this is a level of semantic structuring that cannot be gauged in the NL responses.

These parallels suggest that patterns of successes and errors in our observed outputs are not simple idiosyncrasies of instruction or output format, but may indicate deeper patterns of strength and limitation in the models' capacity for semantic analysis.

# F    Automated format fixes

Our automated format fixes, used for two out of three of our settings for Smatch calculation on GPT-4 few-shot parses, consist of the three simple rule-based fixes detailed below:

1. Keep only first full AMR, and ignore any subsequently generated content (e.g., if model generates multiple separate AMR structures for a single sentence).

2. For retained AMR parse, delete any unmatched right parentheses.

3. Identify duplicates among concept variable names (e.g., "s" in *(s / sphere)*), and replace with non-duplicates.

**"The K-pop music sphere constantly churns out the newest catchy K-pop songs to vie for listeners' attention."**

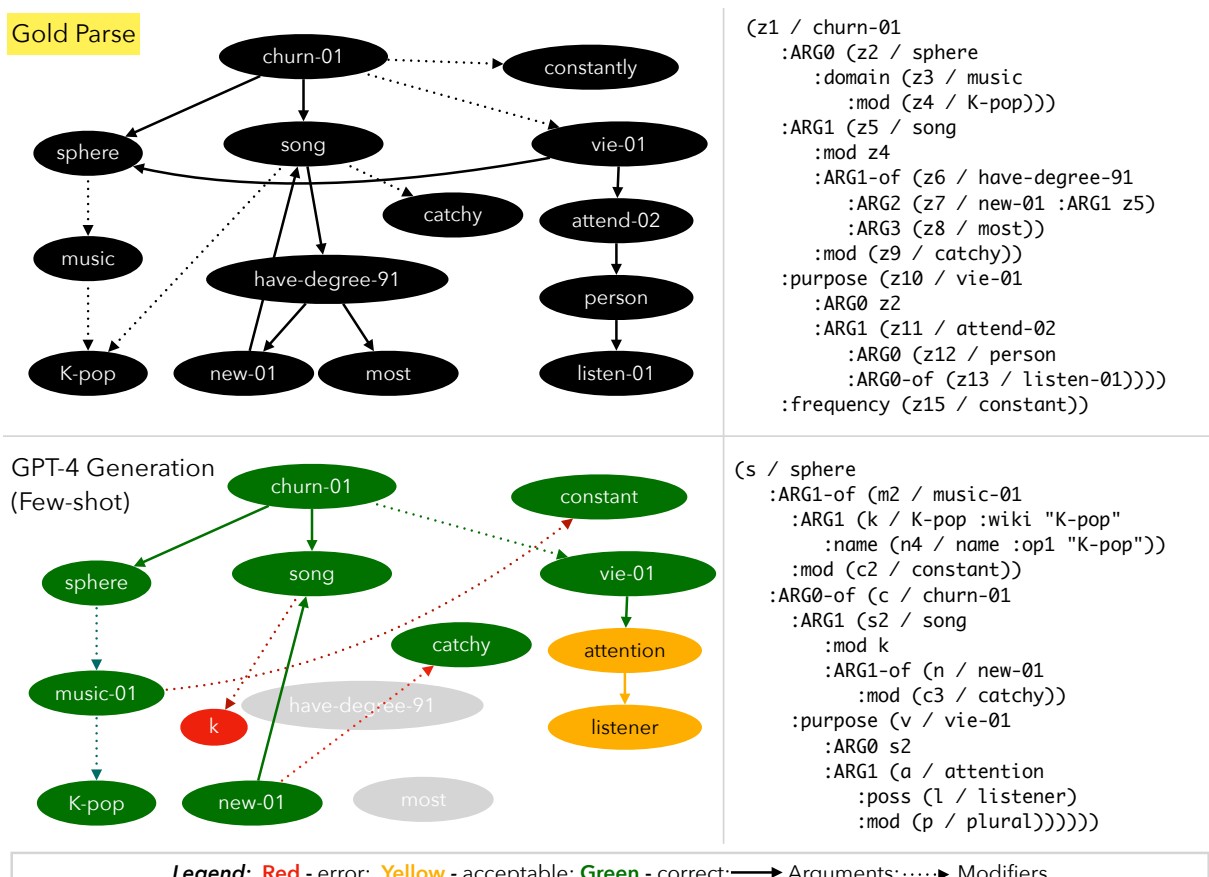

```
(z1 / churn-01
    :ARG0 (z2 / sphere
        :domain (z3 / music
            :mod (z4 / K-pop)))
    :ARG1 (z5 / song
        :mod z4
        :ARG1-of (z6 / have-degree-91
            :ARG2 (z7 / new-01 :ARG1 z5)
            :ARG3 (z8 / most))
        :mod (z9 / catchy))
    :purpose (z10 / vie-01
        :ARG0 z2
        :ARG1 (z11 / attend-02
            :ARG0 (z12 / person
            :ARG0-of (z13 / listen-01))))
    :frequency (z15 / constant))
```

```
(s / sphere
    :ARG1-of (m2 / music-01
        :ARG1 (k / K-pop :wiki "K-pop"
            :name (n4 / name :op1 "K-pop"))
        :mod (c2 / constant))
    :ARG0-of (c / churn-01
        :ARG1 (s2 / song
            :mod k
            :ARG1-of (n / new-01
                :mod (c3 / catchy))
        :purpose (v / vie-01
            :ARG0 s2
            :ARG1 (a / attention
                :poss (l / listener)
                :mod (p / plural))))))
```

*Legend:* **Red** - error; **Yellow** - acceptable; **Green** - correct; ⟶ Arguments; ┈┈▶ Modifiers

(a) Parsing generations vs. Gold parse

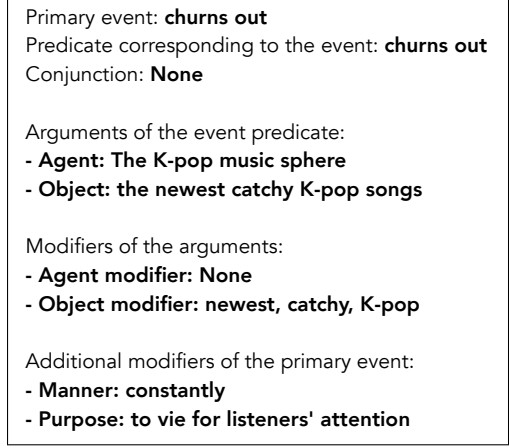

(b) NL Response

Figure 5: A mostly good example

**"There was no issue as we boarded the train, and**
**we sat down and put the luggage next to us."**

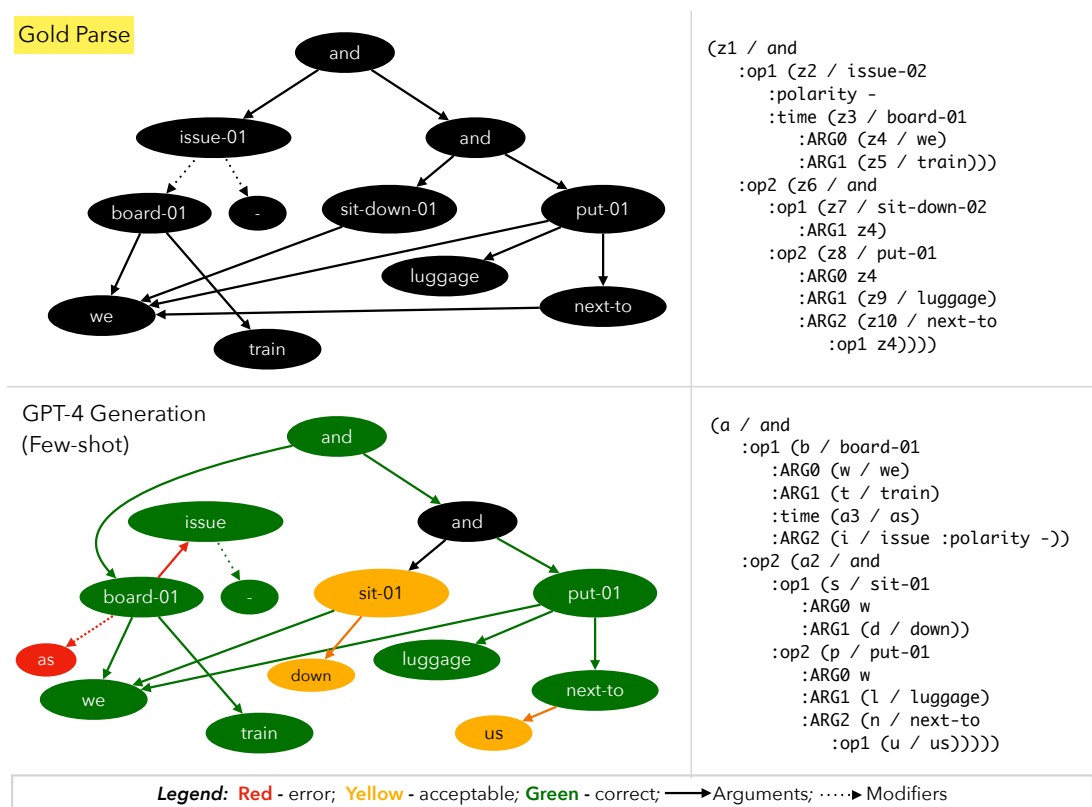

*Legend:* **Red** - error; **Yellow** - acceptable; **Green** - correct; ——▶ Arguments; ·····▶ Modifiers

(a) Parsing generations vs. Gold parse

Primary Events: **There are two equally primary events connected by the conjunction "and".**

2. Primary Event 2: **"we sat down"** and
1. Primary Event 1: **"we boarded the train"**            **"put the luggage next to us"**

Predicate corresponding to event 2 (a): **"sat down"**
Predicate corresponding to event 1: **"boarded"**    Arguments of the event predicate (a):
Arguments of the event predicate:            - **Subject: "we"**
  - **Subject: "we"**                    Modifiers of the arguments (a): **None**
  - **Object: "the train"**
Modifiers of the arguments: **None**

Predicate corresponding to event 2 (b): **"put"**
Arguments of the event predicate (b):
Additional Modifiers of the event:            - **Subject: "we" (the same "we" as "sat down")**
**"There was no issue as"**                - **Object: "the luggage"**
                        Modifiers of the arguments (b):
(This modifier indicates that the boarding        - **Modifier of Object: "next to us"**
happened without any problem.)

Additional Modifiers of the event (a) and (b): **None**

(b) NL Response

Figure 6: A semi-good example

**"Thus you can imagine my amazement, at sunrise, when I was awakened by an odd little voice."**

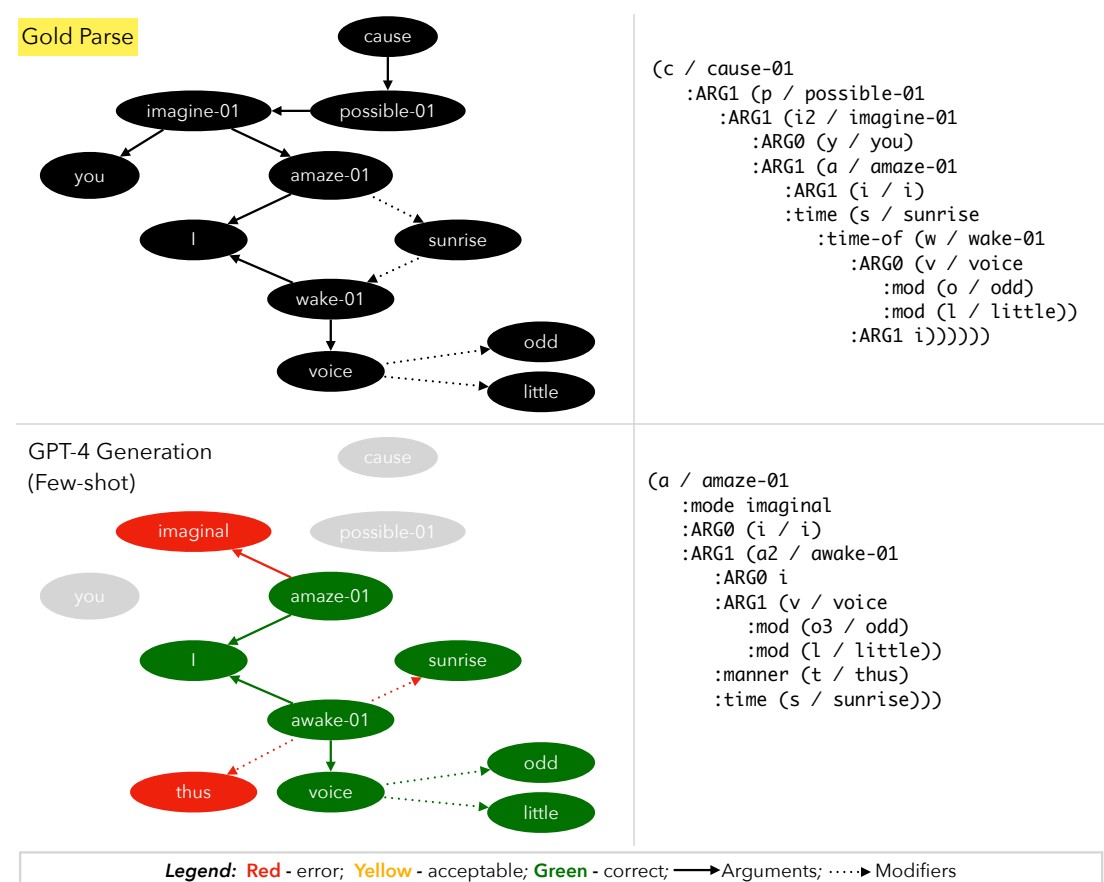

(a) Parsing generations vs. Gold parse

Primary event: **"I was awakened"**
Predicate corresponding to the event: **"was awakened"**

Arguments of the event predicate:
- **Subject: "I"**
- **Object: "by an odd little voice"**

Modifiers of the arguments:
- **Subject: No modifiers**
- **Object: "odd" (modifying "little voice"), "little" (modifying "voice")**

Additional modifiers of the primary event:
- **Time: "at sunrise"**
- **Reason: "Thus, you can imagine my amazement"**

(b) NL Response

Figure 7: A bad example

**"With delicate crochet flower appliques throughout this top, it is a standout piece that will look wonderful either with the matching crochet skirt or paired with simple denim shorts to let the top shine."**

Primary event: "**look wonderful**"
Predicate: "**look**"

Arguments: "**it**" (referring to the top),
"either with the matching crochet skirt or paired with simple denim shorts"

Modifiers of arguments: "**delicate crochet flower appliques throughout**" (modifying "it"/the top)

Additional modifiers of the event: "**standout piece**" (modifying the manner in which the top will look wonderful), "**to let the top shine**" (modifying the purpose)

(a) NL Response

```
(z1 / piece
    :domain (z2 / it)
    :ARG0-of (z3 / look-02
        :ARG1 (z4 / wonderful-03)
        :manner (z5 / or
            :op1 (z6 / pair-01
                :ARG1 z2
                :ARG2 (z7 / skirt
                    :mod (z8 / crochet)
                    :ARG1-of (z9 / match-01)))
            :op2 (z10 / pair-01
                :ARG1 z2
                :ARG2 (z11 / shorts
                    :mod (z12 / denim)
                    :ARG1-of (z13 / simple-02))
                :purpose (z14 / let-01
                    :ARG0 z6
                    :ARG1 (z15 / shine-01
                        :ARG1 (z16 / top
                            :mod (z17 / this))))))))
    :mod (z18 / standout))
```

(b) Gold Parse

Figure 8: Another bad example from NL responses