# OpenReview forum: ""You Are An Expert Linguistic Annotator": Limits of LLMs as Analyzers of Abstract Meaning Representation"
_EMNLP/2023/Conference — EMNLP 2023 Findings_

### Official Review · Reviewer_4W3E · 2023-07-31

**Soundness:** 3

**Excitement:**

3: Ambivalent: It has merits (e.g., it reports state-of-the-art results, the idea is nice), but there are key weaknesses (e.g., it describes incremental work), and it can significantly benefit from another round of revision. However, I won't object to accepting it if my co-reviewers champion it.

**Paper Topic And Main Contributions:**

The paper presents an analysis of the GPT-3, GPT-4, and ChatGPT focusing on Abstract Meaning Representation formalism. The analysis investigates if these tools can serve as an expert linguistic annotator for AMR. This investigation analyzed two settings: the direct production of AMR structures based on examples of zero- and few-shot and the indirect partial reconstruction of AMR via metalinguistic natural language queries.

The paper is well-writtten, structured, and easy to follow. The main contribution is a qualitative analysis of the ability of language models to generate AMR structures from zero- and few-shot settings. This investigation is interesting, as language models may be used to produce AMR corpora for low-resource languages. However, the paper concludes the analyzed models have virtually 0% success in producing fully accurate parses. From this conclusion, I was curious, how was the acceptability of the generated AMR graph evaluated?

Although the generated ARM graphs have errors, is not it easier to edit a partial AMR graph than annotate one from scratch?

Regarding the errors produced by the language models, is there a pattern of errors produced by them? Is it possible to apply post-processing to solve or mitigate them?

Finally, in addition to the performed analysis, I expected to see a comparison between the generated graph by the language models and the gold graphs using an AMR metric.

**Questions For The Authors:**

1) How was the acceptability of the generated AMR graph evaluated?

2) Although the generated ARM graphs have errors, is not it easier to edit a partial AMR graph than annotate one from scratch?

3) Regarding the errors produced by the language models, is there a pattern of errors produced by them?

4) Is it possible to apply post-processing to solve or mitigate them?

**Reasons To Accept:**

- Paper well-written, strucured, and easy to follow

- Paper presents an interesting qualitative analysis of the ability of language models to produce AMR structures

**Reasons To Reject:**

- Lack of quantitative analysis to check the accuracy of language models

- Lack of detail in the qualitative analysis to conclude the acceptability of language models

**Reproducibility:**

N/A: Doesn't apply, since the paper does not include empirical results.

**Reviewer Confidence:**

4: Quite sure. I tried to check the important points carefully. It's unlikely, though conceivable, that I missed something that should affect my ratings.

---

> ### Author Rebuttal · Authors · 2023-08-28
>
> We thank the reviewer for their thoughtful assessment and useful questions. Below we address the questions in turn.
>
> **On acceptability metric details.**
>
> For our overall acceptability measure, an AMR expert among the authors assessed whether each parse could be a valid representation of the sentence meaning, based on the AMR annotation guidelines, regardless of match to the gold annotation. This was intended to give fairer credit to model outputs – none of the models’ generated parses managed to perfectly match the gold annotation, but it was possible that some parses may still accurately represent the meaning of the sentence, with some annotation differences from the gold parse. So this measure used expert assessment to judge parse validity in this broader sense. These assessments also generally forgave minor structural/formatting errors, as long as a correct semantics could be interpreted.
>
> We thank the reviewer for flagging this confusion, and we will make this description clearer in the revised version.
>
> **On making comparisons using an AMR metric.**
>
> Thank you for raising this important suggestion. As we argue in the submission, our aim here was to do finer-grained manual evaluation, to gain insights into model success/failures not available from simple larger-scale quantitative evaluations like smatch or SemBLEU. However, we agree that inclusion of a more standard AMR metric like smatch will be helpful to contextualize and support our claims! We provide those results here, and we will happily add them to the revised version of the paper.
>
> First, baseline smatch scores for our test sets from the supervised AMRBART parser.
>
> Smatch – AMRBART:
>
> - AMR3: 0.78
> - 2023: 0.79
> - LPP: 0.75
>
> Next, smatch scores for our reported GPT-4 outputs in the few-shot setting (the best output quality we found). Note that the model’s generated parses are often flawed to the point of smatch not being able to run, so we report three methods for obtaining scores:
>
> - [The scores in square brackets] reflect automated format fixes, then manually fix remaining formatting errors as needed.
> - For (the scores in parentheses), we apply several automated format fixes, then for parses that still fail, replace them with single-node placeholder parse.
> - For {the scores in curly brackets}, we use no automated format fixes, and replace all failing parses with single-node placeholder parses
>
> Smatch – GPT-4 few-shot:
>
> - AMR3 : [ 0.51 ] ( 0.48 ) { 0.42 }
> - 2023 : [ 0.47 ] ( 0.40 ) { 0.26 }
> - LPP : [ 0.51 ] ( 0.47 ) { 0.41 }
>
> It is clear that, based on these metrics, GPT-4 output quality is far below that of the supervised AMR parser. We will be happy to add these results to the revised version, to further support our points while contextualizing with respect to a standard AMR metric. We appreciate the reviewer making this suggestion.
>
> **On being easier to edit partial AMRs rather than write from scratch.**
>
> Certainly! This is a good observation. However, this is somewhat orthogonal to the point of our paper – we aren’t arguing that these noisy/flawed AMR parses could not have any use. If desired, existing supervised AMR parsers can already enable that type of parse-then-revise workflow (more effectively, based on smatch above). Our point here is more focused on analyzing and reporting LLMs’ capabilities in this domain.
>
> **On error patterns and post-processing to mitigate errors.**
>
> Though we can make some generalizations about the errors that we observe in model generations, the fact is that the errors are quite diverse and numerous – so while we certainly expect post-processing to be able to improve the parses, a key challenge is that the necessary fixes tend to be fairly sentence- or parse-specific. To help in illustrating this, and in response to other helpful comments indicating that having more examples would be beneficial, we are including a number of examples with discussion – we have included these in the response to Reviewer xh4S above. Please see that response for some examples to illustrate the kind of variation and idiosyncrasy we observe in these generations and their errors.

---

### Official Review · Reviewer_xh4S · 2023-08-02

**Soundness:** 3

**Excitement:**

3: Ambivalent: It has merits (e.g., it reports state-of-the-art results, the idea is nice), but there are key weaknesses (e.g., it describes incremental work), and it can significantly benefit from another round of revision. However, I won't object to accepting it if my co-reviewers champion it.

**Missing References:**

References to AMR evaluation, as noted above

**Paper Topic And Main Contributions:**

This paper investigates the capability of GPT-3, ChatGPT, and GPT-4 to produce AMR parses for 30 test sentences, using both zero-shot and five-shot scenarios.  Four evaluation criteria are used in all cases, with an additional four for the five-shot evaluations.  The main contribution is showing that while the LLMs can generate some aspects of a gold AMR parse, it also frequently makes errors of various types.   An interesting aspect of the paper (although in the appendix) is that they also included recent text, for which they created the gold AMR parses, to check whether the LLM's had been learning from earlier officially released material.  They found no significant (informally speaking) difference with such cases.

**Questions For The Authors:**

update to the below.

Regarding "We intend here to investigate the capabilities that have been infused in LLMs via the pre-training process", isn't it kind of strange to have to ask this question w/o  being able to know if the LLMs were trained on AMR material to begin with?  Suppose GPT4 or ChatGPT or whatever is trained on AMR data in some new iteration.  I could imagine someone then writing a paper saying "big improvement in ChatGPT - it can now produce AMR parses almost as well as a human annotator".  But I'm not sure that would be meaningful in any way.

In general, thanks for the detailed response. I am upgrading my soundness and reproducibility scores.

-----------------------------------------------------------------------------------------------
A. The paper asks if LLMs can serve as "an expert linguistic annotator"?   But an "expert linguistic annotator" is not just someone with a fluent command of the language, but it also requires training in the annotation guidelines, etc.    While the LLMs seem to have knowledge, to some extent of the format of AMR, the don't seem to know a lot about AMR.   So perhaps it's doing as well as a typical person would do w/o knowing anything about AMR?  What would happen if instead of 5-shot, you just feed it all the training data, as if training an AMR parser.  One would hope that it could at least learn as well as a specialized parser.  But maybe not?

B. The AMR events seem to encode some amount of word sense disambiguation by referring to particular propbank entries, such as "churn-01".  If my understanding of that is correct, did your criteria require getting the number correct?  Were there any cases where it got the wrong number for a word, and so was incorrect for that reason?

**Reasons To Accept:**

It is a contribution to the question of what LLMs are capable of, in this case by querying how well they can simulate the role of a annotator.

Update: Another contribution is the definition of the evaluation framework.

**Reasons To Reject:**

Update to the below: The authors do a good job in their response of clarifying the evaluation criteria.  The examples should go into an appendix.  I am uncertain though whether this is really a short paper.  The appendix already has a good amount of material related to the evaluation criteria, which is not a secondary aspect of the paper - it's one of its contributions.

I am somewhat concerned by the authors' comments "We also very much appreciate the reviewer’s suggestion to further bin the errors into categories – this indeed would be ideal. However, what we have found is that making further generalizations at lower levels of granularity is not very feasible because the models’ errors are quite diverse and sentence-specific."    To me this indicates that it's really just too little data, if even a manual evaluation could not categorize the errors in some more abstract way.

I really appreciated the smatch comparison with AMRBART - quite a contrast!

----------------------------------
1. The evaluation of the AMR parses seems very ad-hoc. I am not very familiar with AMR work, but just a quick check turns up work on evaluation metrics for AMR (e.g. SEMBLEU, Smatch).  Of course, with only 30 sentences, a quantitative measure using those may not be very revealing.  The authors state that they "prioritize close, fine-grained analyses of the successes and errors of the models in capturing these meaning structures, in order to provide detailed insights into model capabilities."  But it is very hard to actually get an idea of what the LLMs are getting wrong from the criteria they give.  It gets the top node wrong 20-40%, "reflecting routine failures to incorporate clausal and discourse relations in the top node" - such as what?  There is one example in the main text, and three more in the appendix, but with only 30 sentences, and the promise of "fine-grained analyses", it needs some sort of better accounting of what it's getting wrong - e.g. # of sentences due to coordination mistake, # to due to subordinate/main clause confusion, etc.   So the paper is suggestive, but not quite there yet.

2. Connected to that, the description of the criteria is confusing.  The paragraph at lines 172-183 is supposed to correspond, I think, to the columns in Table 1.   Does (178-9) "select the correct event as the main event concept" mean just "select the main event concept correctly"?   This is a "more relaxed criterion" (202-203)  but it's not obvious why. The appendix describes this as "We evaluate whether the main event of the concept is surfaced as the top event concept in the tree", which I was also confused by.  I think a couple of full examples would really help - e.g. here is an example of a correct main event relation but incorrect top node.

**Reproducibility:**

3: Could reproduce the results with some difficulty. The settings of parameters are underspecified or subjectively determined; the training/evaluation data are not widely available.

**Reviewer Confidence:**

3: Pretty sure, but there's a chance I missed something. Although I have a good feel for this area in general, I did not carefully check the paper's details, e.g., the math, experimental design, or novelty.

**Typos Grammar Style And Presentation Improvements:**

generally good aside from the comments above.  Problem with the sentence at 544-547 - "Because *an* AMR parse ...valid parse*s*"

---

> ### Author Rebuttal · Authors · 2023-08-28
>
> We thank the reviewer for their careful assessment and detailed questions. Below we address individual points and inquiries.
>
> **On the lack of standard metrics like smatch.**
>
> We appreciate the reviewer raising this point. As we argue in the submission, our aim here was to do finer-grained manual evaluation, to gain insights into model success/failures not available from simple larger-scale quantitative evaluations like smatch or SemBLEU. That said, we agree that inclusion of a more standard AMR metric like smatch will be a helpful addition to contextualize and support our claims! So we provide those numbers here for the reviewer’s reference, and we will happily add them to the revised version of the paper.
>
> First, baseline smatch scores for our test sets from the supervised AMRBART parser.
>
> Smatch – AMRBART:
>
> - AMR3: 0.78
> - 2023: 0.79
> - LPP: 0.75
>
> Next, smatch scores for our reported GPT-4 outputs in the few-shot setting (the best output quality we found). Note that the model’s generated parses are often flawed to the point of smatch not being able to run, so we report three methods for obtaining scores:
>
> - [The scores in square brackets] reflect automated format fixes, then manually fix remaining formatting errors as needed.
> - For (the scores in parentheses), we apply several automated format fixes, then for parses that still fail, replace them with single-node placeholder parse.
> - For {the scores in curly brackets}, we use no automated format fixes, and replace all failing parses with single-node placeholder parses
>
> Smatch – GPT-4 few-shot:
>
> - AMR3 : [ 0.51 ] ( 0.48 ) { 0.42 }
> - 2023 : [ 0.47 ] ( 0.40 ) { 0.26 }
> - LPP : [ 0.51 ] ( 0.47 ) { 0.41 }
>
> It is clear that, based on these metrics, GPT-4 output quality is far below that of the supervised AMR parser. This helps to contribute to our overall point, while contextualizing with respect to a standard AMR metric, so we will be happy to add these results to the revised version. We appreciate the reviewer making this suggestion.
>
> **On confusing criterion descriptions.**
>
> Thank you for pointing out this confusion. The top node in an AMR parse is not always an event concept. Concepts expressed by conjunction, discourse relations and modality can take the top node position in an AMR parse. For example, in a sentence like “However, they protested …”, “have-concession-91” would take the top node position above the “protest-01” node. The “TopNode” (L180) measure assesses whether the top node is correct (i.e., concept of have-concession-91 is accurate). The “MainRel” (L178) measure assesses whether the main event of the sentence surfaces as the top event relation (disregarding any potential non-eventive relations; e.g., credit is given if the concept “protest” does appear as the top event node in the tree). Therefore, MainRel is a more relaxed measure.  We will clarify these details in the updated version of the paper.
>
> **On model error analysis.**
>
> We hope that the clarification above also helps partly in clearing up what categories of errors are indicated by penalties in each of our measures. We also very much appreciate the reviewer’s suggestion to further bin the errors into categories – this indeed would be ideal. However, what we have found is that making further generalizations at lower levels of granularity is not very feasible because the models’ errors are quite diverse and sentence-specific. We completely agree, however, that it would be beneficial to add more examples to help in showing what this variation looks like, so we provide a couple of illustrative examples here (please see below), and we will happily add these to the main paper and/or appendix in the revised version.
>
> **On providing more training data to LLMs.**
>
> This is a good observation, and we agree that providing more training data has the potential to improve LLMs’ generation of AMRs. If the reviewer has in mind providing full training sets within the in-context learning setting, there is the practical limitation that inputting entire training datasets as in-context examples is not feasible due to context-length limits.
>
> If the reviewer has in mind simply fine-tuning the models on AMR generation, we can assume that this would indeed have a positive effect on the generation quality. However, that type of experiment is somewhat outside the scope of our current submission, as the purpose of this submission is specifically to assess the knowledge and capabilities that these LLMs have for AMR generation *out of the box*, given zero-shot or few-shot in-context learning settings. We intend here to investigate the capabilities that have been infused in LLMs via the pre-training process, so investigating the impacts of fine-tuning constitutes a rather different (albeit also very interesting) research question.
>
> **On sense number.**
>
> No, we never penalize the models for mismatch in sense number. We also do not penalize for generating the incorrect ARG number (e.g., “ARG0” vs “ARG1”) for argument counts (Table 2).
>
>
> **Model error examples. (continued from above)**
>
> > **Example 1**: *“Thus you can imagine my amazement , at sunrise , when I was awakened by an odd little voice”* (LPP sentence)
>
> Discussion: We see below that the two generations diverge wildly from the gold parse. Across the board GPT4 picks up that “amaze” is an important relation, but it fails to appropriately capture that the top event here is “imagine” (among other problems). The few-shot parse receives 1 for Basic form, and 0 for TopNode, MainRel, and Accept measures.
>
> - GPT4 Zero-shot.
>
> ```
> (s / sunrise
>        :manner (a / amaze-01
>                 : experiencer (i2 / imagine-01
>                               :ARG0 (y2 / you)
>                               :ARG1 (i / I)
>                               :time (w / wake_up-01
>                                           :ARG0 (i)
>                                           :ARG1 (v / voice-01
>                                                       :mod (o / odd)
>                                                       :degree (l / little)
>                                                        :time-in (s / sunrise))))))
> ```
>
> - GPT4 Few-shot.
>
> ```
> (a / amaze-01 :mode imaginal
>       :ARG0 (i / i)
>       :ARG1 (a2 / awake-01
>             :ARG0 i
>             :ARG1 (v / voice
>                   :mod (o3 / odd)
>                   :mod (l / little))
>             :manner (t / thus)
>             :time (s / sunrise)))
> ```
>
> - GOLD AMR
>
> ```
> (c / cause-01
>       :ARG1 (p / possible-01
>             :ARG1 (i2 / imagine-01
>                   :ARG0 (y / you)
>                   :ARG1 (a / amaze-01
>                         :ARG1 (i / i)
>                         :time (s / sunrise
>                               :time-of (w / wake-01
>                                     :ARG0 (v / voice
>                                           :mod (o / odd)
>                                           :mod (l / little))
>                                     :ARG1 i))))))
> ```
>
> > **Example 2**: *“Surgical removal of the entire tumor is almost impossible.”* (2023 sentence)
>
> Discussion: The zero-shot parse below is almost entirely nonsensical and hallucinates the concept survive-01. Few-shot is somewhat improved in that remove-01 now rises correctly as the top event; but the role that surgery-01 and possible-01 play within remove-01 is unspecified. Additionally, remove-01 appears twice in the parse. These errors make it impossible to interpret the parse. The few-shot parse receives 1 for Basic form, 1 for MainRel, and 0 for TopNode and Accept.
>
> - GPT4 Zero-shot.
>
> ```
> (s / survive-01 :polarity -
>       :arg1 (c / cell
>             :mod (c2 / cancer))
>       :arg0 (p2 / person :arg1-of (t / think-01
>                                       :arg1 (g / go-09))))
> ```
>
> - GPT4 Few-shot.
> ```
> (r / remove-01
>       :ARG1 (t / tumor
>             :mod (e / entire)))
>       (s / surgery-01)
>       (p / possible-01 :polarity -
>             :ARG1 (r2 / remove-01))
>       :degree (a / almost))
> ```
>
> - GOLD ANNOTATION
> ```
> (z1 / possible-01
>       :polarity -
>       :ARG1 (z2 / remove-01
>             :ARG1 (z3 / tumor
>                   :mod (z4 / entire))
>             :manner (z5 / surgery-01))
>       :mod (z6 / almost))
> ```
>
>
> > **Example 3**: *“It didn't warrant such a reaction, but he was thoughtless.”* (AMR3)
>
> Discussion: Again, below the zero-shot parse is entirely nonsensical but much improved in the few-shot setting. Even so, the few-shot parse still makes semantically nonsensical errors. For example, the gold parse captures the concept of “thoughtless” by using “think-01” in conjunction with negative polarity; GPT-4’s few-shot parse is both redundant (“think” and “thoughtless” both mentioned) and semantically lacking (negative polarity should accompany think-01). The few-shot parse receives 1 for Basic form, 1 for MainRel, 1 for TopNode, and 0 for Accept.
>
> - GPT4 Zero-shot.
> ```
> (d / did-02
>   :mode "-"
>   :ARG1 (w / warrant-01
>     :ARG1 (r / react-01
>       :ARG0 (h / he))
>     :ARG2 (t / thoughtless)
>     :degree (s / such)) )
> ```
>
> - GPT4 Few-shot.
> ```
> (c / contrast-01
>       :ARG1 (w / warrant-01 :polarity -
>             :ARG1 (r / react-01
>                   :ARG2 (s / such))
>             :ARG2 (i / it))
>       :ARG2 (t / think-01
>             :ARG1 (t2 / thoughtless
>                   :ARG1 (h / he))
> ```
>
> - GOLD AMR
>
> ```
> (c / contrast-01
>       :ARG1 (w / warrant-01 :polarity -
>             :ARG0 (i / it)
>             :ARG1 (t / thing
>                   :ARG2-of (r / react-01)
>                   :mod (s / such)))
>       :ARG2 (t2 / think-01 :polarity -
>             :ARG0 (h / he)))
> ```
>
>
> > **Example 4**: *“Now, with the various "Occupy" movements, it is (or ought to be) obvious that a great many people are less than enchanted with capitalism and its expression in the idustrial (sic) democracies, as well as in the countries into which capitalists from the industrial democracies go to invest.”* (AMR3)
>
> Discussion: This example below from ChatGPT zero shot demonstrates an example for badly formatted generation. This parse receives 0 across all measures:
> ```
> [ [now], [with], [ [the], [various], ["Occupy"], [movements] ],
>  [it], [be], [ (or] [ought], [to], [be) ],
>  [ [obvious], [ [that], [ [a], [great], [many], [people] ],
>    [ [ [are], [less], [than], [enchanted] ],
>      [with], [capitalism], [and],
>      [ [its], [expression],
>        [ [in], [the], [industrial], [democracies] ],
>        [ [as], [well], [as],
>          [ [in], [the], [countries],
>            [ [into], [which], [capitalists], [from],
>              [the], [industrial], [democracies],
>              [go], [to], [invest] ] ] ] ] ] ] ] ]
> ```

---

### Official Review · Reviewer_ZiDs · 2023-08-11

**Soundness:** 3

**Excitement:**

3: Ambivalent: It has merits (e.g., it reports state-of-the-art results, the idea is nice), but there are key weaknesses (e.g., it describes incremental work), and it can significantly benefit from another round of revision. However, I won't object to accepting it if my co-reviewers champion it.

**Paper Topic And Main Contributions:**

The paper explores the question of whether LLMs have sufficient linguistic knowledge about the language by analyzing the successes and limitations of GPT-3, ChatGPT, and GPT-4 on the AMR task. The contributions of the paper are as follows:

The paper presents a comprehensive evaluation framework to assess the quality of AMR parses generated by LLMs. This framework includes high-level and low-level criteria for analyzing the structure, validity, and accuracy of the parses.

The paper provides analysis of LLMs' parse results with different prompts and conducts instance analysis and discussions, thereby analyzing the capabilities and limitations of LLMs in performing the AMR task. This analysis helps readers understand the linguistic abilities of the LLMs mentioned in the paper.

**Reasons To Accept:**

The strengths and main benefits of accepting this paper are as follow:
	1. The paper introduces a comprehensive evaluation framework for assessing the quality of Abstract Meaning Representation (AMR) parses generated by large language models (LLMs). This framework provides high-level and low-level criteria, enabling researchers to evaluate and compare the performance of various LLMs on the AMR task.
2. The paper offers insights into the linguistic knowledge and capabilities of LLMs, specifically GPT-3, ChatGPT, and GPT-4, in the context of the AMR task. By analyzing the successes and limitations of these models, the paper sheds light on the strengths and weaknesses of current LLMs in understanding and representing abstract semantics.


**Reasons To Reject:**

The weakness of this paper are as follow:
	1.  The paper utilizes a small test set, consisting of only 10 sentences for each dataset. This limited sample size raises concerns about the potential bias in the results. A larger and more diverse test set may provide a more robust evaluation of the LLMs' performance and help mitigate the impact of any dataset-specific biases.

 The prompts used in the paper are deemed relatively simple. A more sophisticated and refined prompt design could potentially yield different results. Exploring more nuanced and diverse prompts might provide deeper insights into the capabilities and limitations of LLMs in AMR parsing.

**Reproducibility:**

3: Could reproduce the results with some difficulty. The settings of parameters are underspecified or subjectively determined; the training/evaluation data are not widely available.

**Reviewer Confidence:**

3: Pretty sure, but there's a chance I missed something. Although I have a good feel for this area in general, I did not carefully check the paper's details, e.g., the math, experimental design, or novelty.

---

> ### Author Rebuttal · Authors · 2023-08-28
>
> We thank the reviewer for their detailed and thoughtful comments. Below we address the major points.
>
> **On test set size; concerns about bias.**
>
> We appreciate the reviewer’s attention to this concern. As we argue in the submission, our aim here was to do finer-grained manual evaluation, to gain insights into model success/failures not available from simple larger-scale quantitative evaluations like smatch or SemBLEU. The trade-off is that we must work with smaller datasets, and there is a bit more subjectivity in the analysis. This choice aligns with the established tradition of qualitative error analysis on small numbers of examples – and we argue that our approach is well in keeping with recent precedent of qualitatively analyzing small numbers of examples to draw insights about model capabilities (e.g., [1]). This trend is influenced in part both by cost and by complexity of outputs, as are we.
>
> So we absolutely agree with the importance of larger-scale quantitative evaluation, but we believe that there is also an important complementary place for smaller-scale fine-grained evaluation, and here we aim to contribute on that dimension.
>
> We will also note that we were careful in our sampling for our test sets, exerting controlled variation in terms of domain, length, and date of publication, to help to minimize potential biases. So while there is still non-zero risk of bias, we hope to minimize that risk while also taking advantage of the insights to be gained from close, fine-grained analysis.
>
> **On simplicity of prompts.**
>
> We agree that more detailed prompt engineering could influence the results. We believe, however, that there is a strong argument as follows: if a prompt is sufficiently clear, then a failure to perform the task is simply a failure – reliance on particular prompt phrasing or structure is an indication of brittleness in model capabilities. So rather than trying to carefully engineer our prompts to accommodate model brittleness, our goal here is simply to design clear and fair prompts, and then assess model abilities on the task that those prompts describe.
>
> [1] Bubeck, Sébastien, et al. "Sparks of artificial general intelligence: Early experiments with GPT-4." arXiv preprint arXiv:2303.12712 (2023).

---

### Meta-Review · Area_Chair_JZ3p · 2023-09-24

**Recommendation:** 4

**Metareview:**

The paper studies the LLMs’ linguistic knowledge in serving as an expert linguistic annotator. In particular, the paper analyzes the capabilities of the GPT-3, ChatGPT, and GPT-4 models in generating
the Abstract Meaning Representation (AMR) of sentences.

Pros:
* All reviewers have found that the paper is very well-written and easy to follow.
* The paper presents a thorough analysis of the quality of generated AMRs from  GPT-3, ChatGPT, and GPT-4, assessing the strengths and weaknesses of current LLMs.
* The finding that “these models out-of-the-box can accurately identify some core aspects of semantic structure, but there remain key limitations in their ability to support fully accurate semantic analyses or parses” will be valuable to the practitioner interested in semantic parsing and analysis of texts.

Cons:
* The reviewers have rightly raised concerns that the analysis has been done with a fairly small dataset (10 sentences per dataset), and this could lead to bias in the results. There is a lack of large-scale quantitative analysis in the paper.
* * I find the authors’ rebuttal to be convincing that we should allow space for smaller-scale yet comprehensive fine-grained evaluations at ACL venues. As a short paper, the analysis presented in the paper accounts for a solid contribution and will be valuable to the community.
* Other concerns, such as simplicity of prompts etc, were addressed adequately in the authors response.

---

### Decision · Program_Chairs · 2023-10-07

**Decision:**

Accept-Findings

**Comment:**

The paper studies the LLMs’ linguistic knowledge in serving as an expert linguistic annotator. In particular, the paper analyzes the capabilities of the GPT-3, ChatGPT, and GPT-4 models in generating
the Abstract Meaning Representation (AMR) of sentences.

Pros:
* All reviewers have found that the paper is very well-written and easy to follow.
* The paper presents a thorough analysis of the quality of generated AMRs from  GPT-3, ChatGPT, and GPT-4, assessing the strengths and weaknesses of current LLMs.
* The finding that “these models out-of-the-box can accurately identify some core aspects of semantic structure, but there remain key limitations in their ability to support fully accurate semantic analyses or parses” will be valuable to the practitioner interested in semantic parsing and analysis of texts.

Cons:
* The reviewers have rightly raised concerns that the analysis has been done with a fairly small dataset (10 sentences per dataset), and this could lead to bias in the results. There is a lack of large-scale quantitative analysis in the paper.
* * I find the authors’ rebuttal to be convincing that we should allow space for smaller-scale yet comprehensive fine-grained evaluations at ACL venues. As a short paper, the analysis presented in the paper accounts for a solid contribution and will be valuable to the community.
* Other concerns, such as simplicity of prompts etc, were addressed adequately in the authors response.